# Scissor Bite in Growing Patients: Case Report Treated with Clear Aligners

**DOI:** 10.3390/children10040624

**Published:** 2023-03-27

**Authors:** Teresa Pinho, Sara Gonçalves, Duarte Rocha, Maria Luís Martins

**Affiliations:** 1UNIPRO—Oral Pathology and Rehabilitation Research Unit, University Institute of Health Sciences (IUCS), CESPU, 4585-116 Gandra, Portugal; 2IBMC—Instituto Biologia Molecular e Celular, i3S—Inst. Inovação e Investigação em Saúde, Universidade do Porto, 4200-135 Porto, Portugal

**Keywords:** scissor bite, brodie bite, malocclusion, treatment, correction, therapeutics

## Abstract

Scissor bite (SB.) is a rare malocclusion that is challenging to diagnose and is often associated with a retrognathic mandible and a series of functional and structural abnormalities that negatively affect the patient. This article intends to analyze the treatment approaches applied to growing patients younger than 16 years old, comparing the conventional appliances described in the literature and a clinical case treated with clear aligners with mandibular advancement (MA.). SB is primarily related to skeletal Class I and II, according to Angle classification. In the various cases analyzed, it can also be mentioned as a significant number of cases with SB of dental origin (seven of dental and four of skeletal) in young patients. In children and adolescents who still have growth potential, the therapeutic possibilities are numerous. A comprehensive literature search was manually performed from 2002 until January 2023, in PubMed and BVS databases with the following conjugated keywords: “scissor bite OR brodie bite” AND “malocclusion” AND “treatment OR correction OR therapeutics”. The present case report on a young patient demonstrated the efficiency of the clear aligners with MA to correct an SB, associated with several functional and structural anomalies such as Class II division 1 with an increased overjet and overbite as well as a severe curve of Spee in a hypodivergent biotype.

## 1. Introduction

Scissor bite (SB.) is defined as a buccal displacement of maxillary posterior teeth, with or without contact between the lingual surface of the maxillary lingual cusp and the buccal surface of the mandibular antagonist’s buccal cusp. This malocclusion is caused by an excessive maxillary width and a narrow mandibular alveolar process, although the width of the mandibular base is usually normal [1,2]. The SB does not correct spontaneously and becomes progressively worse by over-eruption as the upper teeth continue to occlude on the buccal surfaces of the lower posterior teeth [3].

It is often associated with several functional and structural anomalies, such as mandibular rotation and/or lateral deviation with an inclination of the occlusal plane, facial asymmetry, reduced occlusal height, and trauma to the dentition with associated problems in the periodontal tissue and alveolar bone, as well as Class II relation [4,5].

In children and teenagers who still have growth potential, the therapeutic possibilities are numerous and therefore a good diagnosis and early treatment planning are fundamental in providing a complete rehabilitation of the patient in terms of function, aesthetics, and long-term stability [6].

A new mandibular advancement (MA) device was implemented by Align Technology™ (San José, CA, USA) on clear aligners to treat growing patients with skeletal Class II. This device has principles that are similar to the Twin-Block appliance that promotes MA with subsequent neuromuscular adaptation and a new position [7]. This stage is often preceded by a pre-MA stage, which consists of tooth alignment in both arches, and only then urges the MA stage. This stage includes aligners composed of two pairs of lateral inclined planes (precision wings), located in the posterior area of aligners and positioned buccally. When the patient occludes, those precision wings determine the MA. The forced forward position of the mandible induces a series of changes, promoting the correction of the mandibular retrusion and enhancing the craniocervical posture of the Class II malocclusion [8]. To date, the scientific literature presents few studies evaluating the skeletal effects caused by the use of the MA with clear aligners [9].

For the literature review, the following keywords and MeSH terms were employed in the search strategy, in the PubMed and VHS database, from 2002 to January 2023: “scissor bite OR brodie bite” AND “malocclusion” AND “treatment OR correction OR therapeutics”.

The aim of the present article is to demonstrate the different treatment approaches for the correction of SB in a young patient with a complex clinical case, ultimately treated with clear aligners and precision wings.

## 2. Case Report

### 2.1. Diagnosis and Etiology

A 12-year-old female child had a severe Class II division 1 malocclusion with an increased overjet (13.6 mm, norm = 2.5 ± 2.5 mm) and overbite (8 mm, norm = 2.5 ± 2.5 mm). She presented a bilateral SB on the right side between teeth 14 to 43 and 17 to 47, and on the left side between teeth 25 to 35, with lower incisors occluded in the palatal mucosa, due to a severe curve of Spee. She also had a lower lip interposition with high upper incisor exposition and severe loss of vertical dimension and lower lip interposition (Figure 1).

The etiology of the malocclusion was primarily dental. Despite the SB condition, the upper arch was constricted, while the lower arch was in a U form, with lingual tipping pronounced on the teeth that were on SB (teeth 35, 43, and 47) (Figure 1). Additionally, despite the anterior diastemas on the lower arch, there was a lack of space for the second premolars with impaction tendency on the right one.

A right molar Class II and left molar Class I could be observed, with a bilateral canine total Class II that was more severe on the right side (Figure 1).

The cephalometric analysis found a Class II at alveolar level (A–B distance = 6.9, norm = 5 ± 1°), associated with an increased overjet (13.6°) with a high dento-alveolar component and facial repercussion with a high mento-labial sulcus. There was a high upper incisors pro-inclination (UI/NA= 43°, norm = 22 ± 2°) and lower incisors pro-inclination (mandibular incisor to mandibular plane angle, IMPA = 96°, norm = 90 ± 3°). Taking into account the high overjet, a skeletal component was also inherent; however, Class I (ANB = 3°, norm = 3 ± 2°) was verified due to the severe hypodivergent biotype (Frankfurt-mandibular plane angle, FMA = 19°, norm = 25 ± 3°) with a high mentum projection. (Figure 2).

### 2.2. Treatment Progress

On the first set of aligners, there was no MA phase because the inclusion criteria for precision wings were not present due to the high curve of Spee and the second lower right premolar impaction tendency (Figure 3, Figure 4, Figure 5 and Figure 6). The patient used 39 of 44 total aligners on the first set of aligners (Figure 3, Figure 4, Figure 5 and Figure 6). Additionally, the first additional aligners (AA.) [10] were planned without the MA phase. Because the second right premolar was microdontic, it was not totally erupted, and this is essential to support the precision wings in the MA phase. Therefore, the intra-arch alignment and the curve of Spee improvement was performed before MA. All 32 of the 32 total aligners were used (Figure 7).

In the second AA, MA was accomplished. In this stage, precision wings were present (with three advancement steps), with the last step planned in the edge-to-edge position. This stage is intended to increase the vertical dimension and advance the mandible (Figure 8 and Figure 9). To ensure that the precision wings are correctly settled, the patient also used Class II elastics (Figure 9). The MA phase had a duration of 1 year. Here, 40 of 51 total aligners (Figure 8 and Figure 9) were used because the MA was effective as planned, and a Class III with an edge-to-edge incisor relation was obtained before the end of this aligner set (Figure 8).

The third AA stage was intended to correct the lateral open bite, which used vertical elastics to improve the vertical dimension (Figure 10 and Figure 11). The Class I relation was established and confirmed in the last AA (Figure 10 and Figure 11).

The fourth and last AA were used only 12 h/day, at night, and changed once a month, only to stabilize, since this is a difficult case with a growing patient. All of the 28 total aligners were used (Figure 12).

The total treatment time was 3 years. Since the beginning, aligners were changed once a week in every stage.

### 2.3. Treatment Results

In the first set of aligners, over the duration of 8 months, the alignment was performed with the SB fully corrected, and the space for the impacted second right premolar was recovered.

The first set of AA was essential to improve the microdontic second premolar position, as well as to continue to level the curve of Spee to enhance the MA phase, which occurred on the second set of AA. The excessive overjet was improved through the ideal during the MA phase, as planned, so only 40 of 51 total aligners were used (Figure 7). The normal overjet was reached, and stability was accomplished on the third AA and then confirmed in the fourth AA (Figure 10 and Figure 12).

At the end of the 3 years of treatment, including one year with aligners used only 12 h/day, at night, and changed once a month, the molar and canine Class I molar relation, as well as the curve of Spee with normal overjet and overbite, were confirmed. There are undoubtedly harmonious dental and facial aesthetics, as is visible on the smile line (Figure 13).

Through a panoramic X-ray, a satisfactory root parallelism was obtained (Figure 14).

At post-treatment cephalometric analysis, the skeletal Class in ANB reduced from 3° to −0.9° (norm = 3 ± 2°), due to the point “A” being more retruded, considering the upper incisor retrusion with lingual root torque. This also explains the reduced values for alveolar maxilo-mandibular relation (A-B distance) from 6.9 to 1.5, norm = 5 ± 1°). The high upper incisor’s pro-inclination was normalized (UI/NA= 43° to 23.6°, norm = 22 ± 2°), but the lower incisor’s pro-inclination was worsened (IMPA = 96° to 105.7°, norm = 90 ± 3°), inherent to the overjet correction with compensation. The severe hypodivergent biotype was improved (FMA = 19° to 15.5°, norm = 25 ± 3°).

General cephalometric superimposition before and after treatment showed an improvement in the vertical dimension as well as in the upper incisor inclination that promotes the improvement on mento-labial sulcus and on facial aesthetics, in spite of the point “A” retrusion (Figure 15).

In the follow-up, 6 months after the treatment end, maintenance of the obtained results can be observed, and panoramic X-ray with lower wisdom teeth extraction (Figure 16).

The present clinical case was compared with the literature review described in Appendix A, which was added according to the author/year of publication, title, sample, etiology, treatment, and conclusion.

The articles are related to the etiology and the appropriate treatment plans for SB in young patients. The distinction between adults and young patients was made, considering the end of sutural growth that occurs simultaneously for the sagittal and transverse planes at an average age of 17 years old [11].

## 3. Discussion

To critically discuss the present clinical case with an SB associated with a severe sagittal dental position, an integrative literature review, with data based on the current knowledge of different treatment approaches for the correction of SB in young patients, was performed in PubMed and BVS databases with the following conjugated keywords: “scissor bite OR brodie bite” AND “malocclusion” AND “treatment OR correction OR therapeutics”, limited to the time from 2002 to January 2023 (Appendix A).

Currently, the prevalence of SB is 1.1% in children [12]. It is challenging to diagnose, especially in mixed dentition, since the patient is usually asymptomatic. In this case, beyond the SB, there was a high aesthetic problem, being that this condition was the first motivation for seeking orthodontic treatment. Considering the severity of the malocclusion, it is known from the outset that these cases are difficult to correct, even with orthodontic-surgical treatment [6,13]. The results obtained in this patient, using an initial approach of Invisalign^®^ aligners to promote teeth alignment and then precision wings as a functional appliance to endorse MA, were compared with those from publications using different approaches in growing patients.

If the malocclusion primarily involves buccolingual tipping of the dentition without severe skeletal asymmetry, various orthodontic treatments such as functional appliances [1,2,6,14,15], intermaxillary crossed elastics combined with occlusal bite tubes or plates [1], and transpalatal arch (TPA) [16] have been suggested as conventional treatment options [17]. To avoid undesirable side effects on teeth, temporary skeletal anchorage devices such as mini-screws (MS) can also be used [17]. However, when a young patient with a SB also has a severe facial asymmetry discrepancy in basal arch width, a surgical approach may be the best option [17].

However, the ideal option would be to treat these cases as soon as possible to avoid skeletal damage that would worsen, with altered function due to the SB condition. The interceptive treatment in dental SB treated with mandibular expansion appliance, intermaxillary cross elastics, and/or occlusal bite tubes is a minimally invasive option that can provide complete rehabilitation with a satisfactory prognosis in hypodivergent facial biotypes [1]. Nevertheless, the treatment with intermaxillary cross elastics requires excellent cooperation and can often produce unwanted molar extrusion, resulting in clockwise rotation of the mandible, inclination of the occlusal plane, occlusal prematurity, or anterior open bite [5,18].

Several functional appliances can be used for the SB treatment, whether of dental or even skeletal etiology, such as Schwarz appliances for the expansion of the mandibular dental arch or the Hyrax expander (in the case in question, working as a compressor) when the maxilla is wide [14]. Additionally, the Herbst appliance was used in an adolescent patient to correct a Class II skeletal relationship by advancing the mandible after unblocking it, allowing it to stay in a more stable transverse relationship. Thus, the mandible’s wider part occludes the maxilla’s narrower region, correcting the SB. Therefore, this functional appliance in the growing age can successfully correct skeletal SB by MA, presenting advantages in a generally short time, between 6 to 8 months (acting 24 h a day), and does not require the patient’s cooperation [6].

On the other hand, the quad-helix constriction appliance to reduce the width of the maxillary arch and elevate the bite, as well as the bi-helix appliance to expand the mandibular arch, were used in the treatment of a skeletal SB on a 9-year-old child, as described by Nojima K. et al. [2]. In that case, treatment in the mixed dentition with the quad helix appliance effectively reduced the maxillary arch width, providing good results in a short period of time and independent of the patient’s cooperation [2]. Nevertheless, the application of a removable, slow maxillary contraction appliance was reported as an inexpensive and effective way to treat transverse maxillary asymmetry [14].

To reduce the need for young patient collaboration, the transforce appliance for mandibular arch widening was also used to treat the dental SB. This appliance, when compared to the modified lingual arch, does not require as frequent activations. Furthermore, since it causes gentle continuous pressure with nickel–titanium springs, patient discomfort is usually minimal. Early treatment by expansion is currently advocated to redirect erupting teeth to their normal positions and eliminate premature occlusal contacts, thus favoring beneficial dentoalveolar changes during growth periods [15].

Currently, orthodontic mini-screws can be used in teenagers as direct or indirect skeletal anchorage devices to correct the SB with minimal side effects [16]. A disadvantage of direct anchorage is that at least two mini-screws are needed, not only to control torque, but also to prevent rotation. Therefore, an indirect anchorage system with only one mini-screw has been developed to overcome the shortcomings of skeletal anchorage [19].

Yun S. et al. [19] reported the treatment of a young patient case with a newly designed spring (dragon-helix) combined with an indirect skeletal anchorage. This spring provides effective tooth movement and the convenience of a simple and short design that results in less injury to the oral tissues. Thus, it is necessary to frequently check its stability [19].

Mini-screws can be used to successfully correct bilateral or unilateral SB and also minimize unnecessary biological reactions such as root resorption. Some authors demonstrated that the use of mini-screws and the transpalatal arch (TPA) with a distal extension of the arm is useful for SB correction because, unlike intermaxillary elastics, it creates an intrusive force along with palatal traction without causing tooth extrusion [5,16].

For young growing patients, mandibular distraction osteogenesis with high bone regeneration potential is a great treatment option [17]. This procedure involves osteotomy and a distractor appliance to stretch gradually. It has demonstrated success in expanding the mandibular basal bone for the correction of the SB. Importantly, distraction osteogenesis is most effective in younger patients with growth potential [3]. Some authors demonstrated in their clinical case that distraction osteogenesis is a very predictable, successful, stable, affordable, and comfortable procedure in the treatment of skeletal SB [20].

Currently, there is a growing demand for more aesthetic treatments among teenagers and adults [21]. In the present case, aligners were used not as an aesthetic device, but as a comfortable and effective approach to unblock the bite in the pre-MA phase with the correction of the dental axes, particularly the negative torque existing in the teeth that were in SB. The mandible blockage from making a natural advancement was due to the hypodivergent biotype associated with the high functional problem of a severe overjet and overbite. The lower incisors were occluded in the palatal mucosa due to a severe curve of Spee, providing atypical swallowing with lower lip interposition and high upper incisor exposition. Additionally, there was a second lower right premolar impaction tendency. To correct this condition and allow for a favorable tooth position to perform the MA phase with precision wings, two sets of aligners were required, considering the growth potential inherent in the hypodivergent facial biotype. Usually, one set of aligners would be enough, but the biological component of the impacted pre-molar made it necessary to carry out two sets [9,22].

The skeletal Class I relationship contradicts the severe Class II relationship existing at a dental level. This can be explained by the fact that the child has a marked loss of the occlusion vertical dimension, with the anterior rotation of the mandible aggravated by the protruded chin. This situation promotes a very concave profile, which, in addition to being considered unsightly, is a typical aged profile [9,22].

The precision wings (used in the MA phase) provided an orthopedic action of MA, also improving the occlusion vertical dimension. Therefore, the MA stage led to a transitional lateral open bite, considering that the existing severe curve of Spee had to be improved by the extrusion of the premolar teeth using intermaxillary elastics, and not so much by the intrusion of the lower incisors, which was corrected in the pre-MA phase. This approach led to a posterior rotation of the mandible with an increase in the occlusion vertical dimension. It allowed for MA without worsening the profile that was already concave, as can be seen in the initial vs. final tracing superimposition [9,22].

The excessive overjet was improved through the ideal during the MA phase, as planned. However, a small relapse was accomplished, considering the virtual coordination of the arches, resulting in a normal sagittal relation with a normal overjet and overbite, as well as harmonious facial aesthetics.

A limited number of articles on the desired content was found within ten years of the selected time period. Therefore, more clinical case reports and, ideally, case studies are needed to better understand the etiology and proper treatment of this malocclusion, which is still difficult to diagnose, mainly because the patients are asymptomatic in most cases.

## 4. Conclusions

The presented clinical case, treated with a new approach, demonstrated the efficiency of the clear aligners in the first stage and then a conjugation with precision wings to perform MA, correcting the severe malocclusion with SB associated with complex functional and structural anomalies such as Class II dental relation and severe curve of Spee in a hypodivergent biotype.

## Figures and Tables

**Figure 1 children-10-00624-f001:**
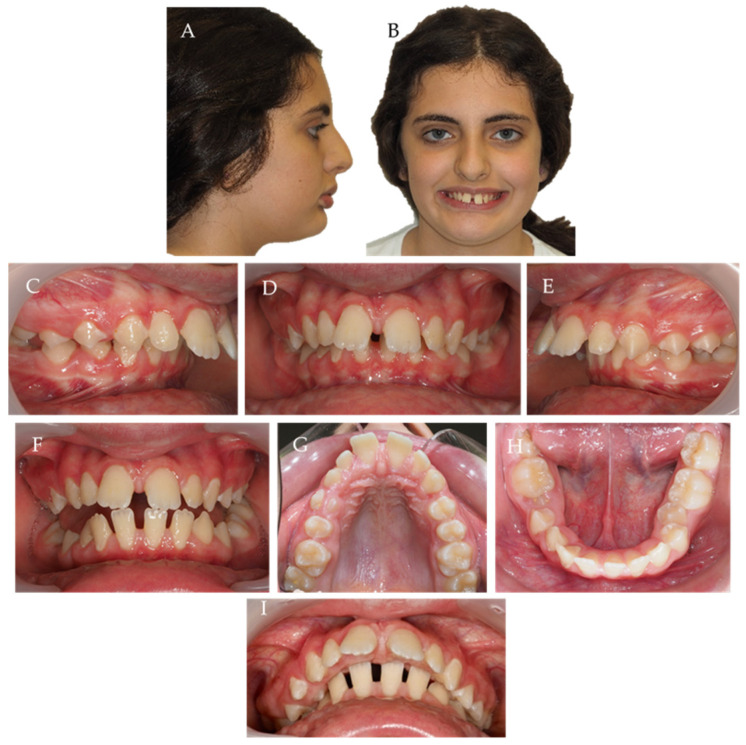
Twelve-year-old female child with a bilateral SB on the right side between teeth 14 to 43 and 17 to 47, and on the left side between teeth 25 to 35, with a severe overjet and overbite. (**A**) profile photo; (**B**) smile; (**C**–**E**) intra-oral photos in maximum intercuspation/centric relation; (**F**) protrusive guide; (**G**,**H**) occlusal photos; (**I**) overjet view.

**Figure 2 children-10-00624-f002:**
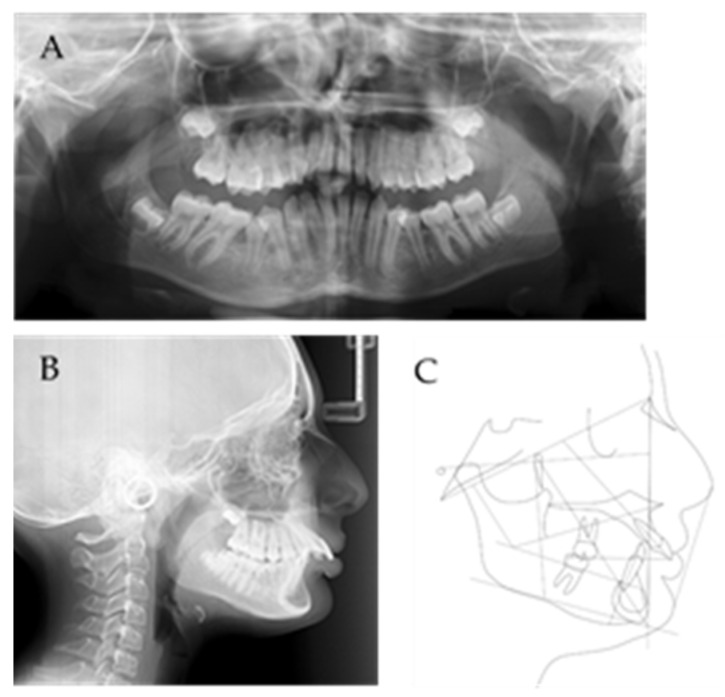
Initial panoramic X-ray (**A**), teleradiograph (**B**) and cephalometry (**C**).

**Figure 3 children-10-00624-f003:**
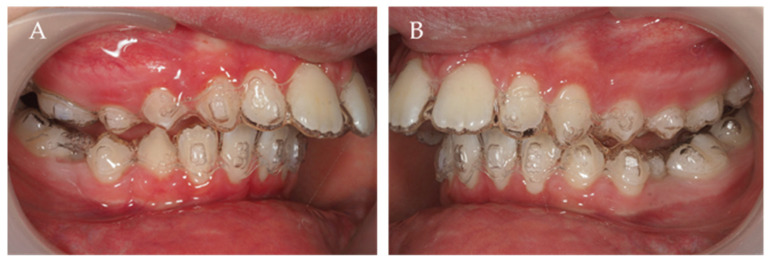
Intra-oral lateral right (**A**) and left (**B**) photos with the first set of aligners.

**Figure 4 children-10-00624-f004:**
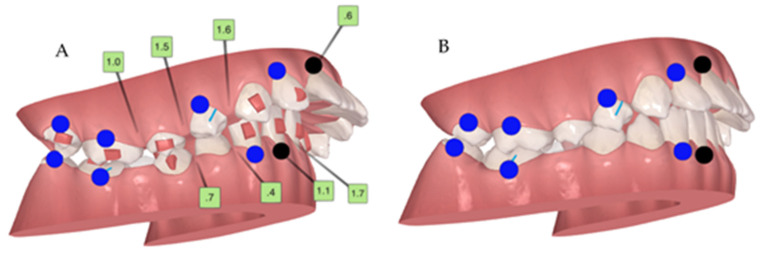
Initial ClinCheck^®^. (**A**) beginning; (**B**) planned. Blue dots—moderate movements; black dots—complex movements.

**Figure 5 children-10-00624-f005:**
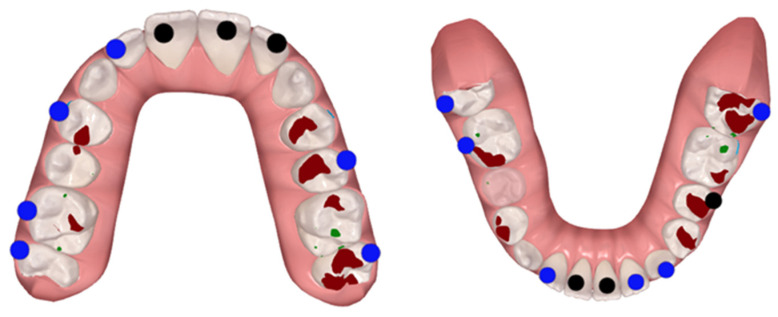
Initial ClinCheck^®^. Planned occlusal contacts. Blue dots—moderate movements; black dots—complex movements.

**Figure 6 children-10-00624-f006:**
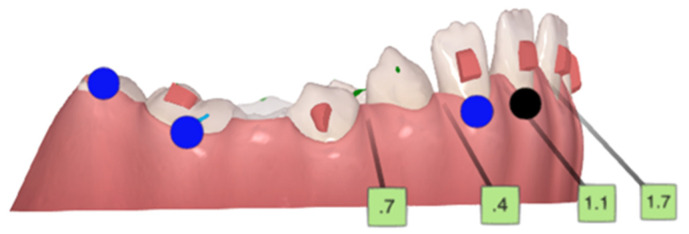
Initial curve of Spee, with eruption compensation on tooth 45. Blue dots—moderate movements; black dots—complex movements.

**Figure 7 children-10-00624-f007:**
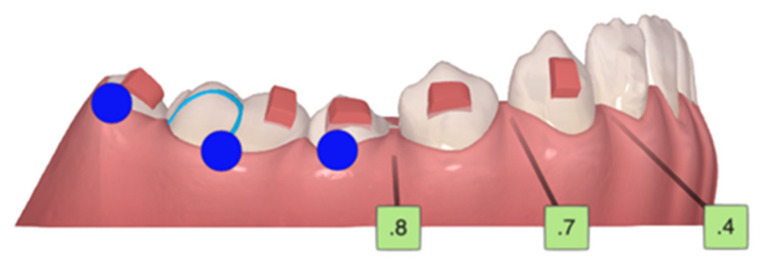
First AA, with tooth 45 microdontic. Blue dots—moderate movements.

**Figure 8 children-10-00624-f008:**
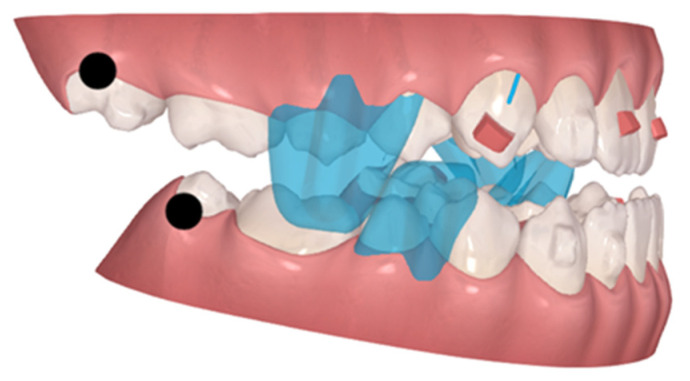
Second AA, with precision wings (blue color) on the last jump. Black dots—complex movements.

**Figure 9 children-10-00624-f009:**
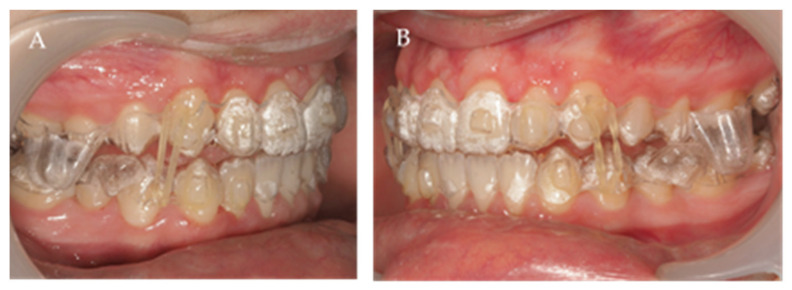
MA stage, with precision wings and Class II elastics on the right (**A**) and left (**B**) side.

**Figure 10 children-10-00624-f010:**
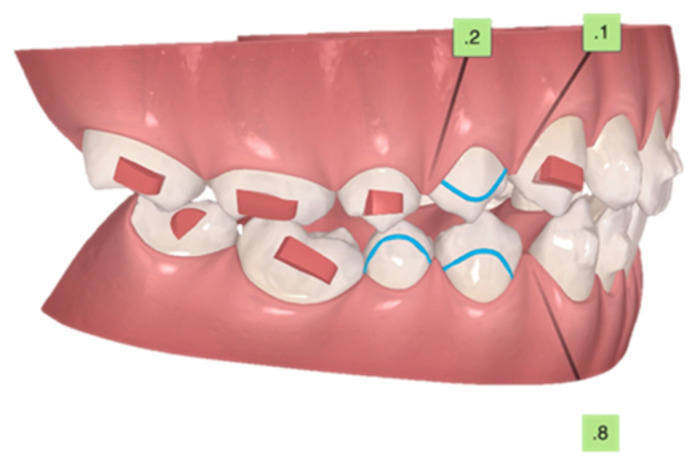
Third AA stage.

**Figure 11 children-10-00624-f011:**
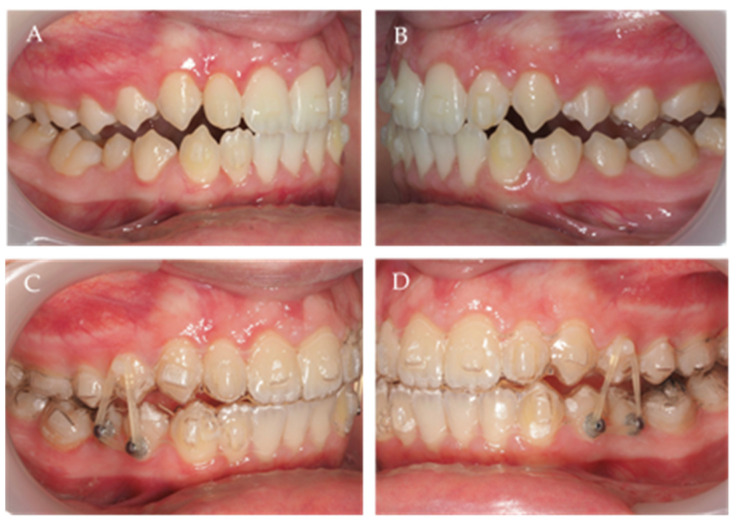
(**A**,**B**) Photographs after second AA and with aligner 1 (**C**,**D**) from third AA stage with vertical elastics.

**Figure 12 children-10-00624-f012:**
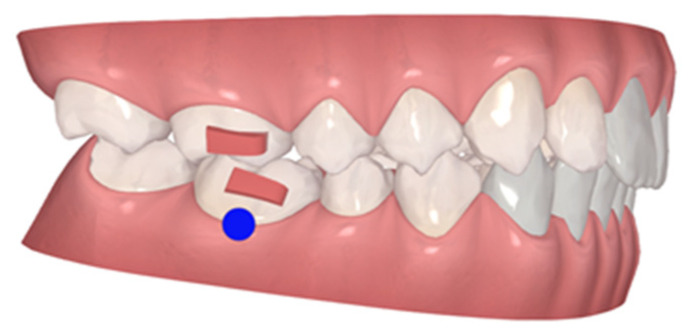
Fourth AA stage. Blue dot—moderate movement.

**Figure 13 children-10-00624-f013:**
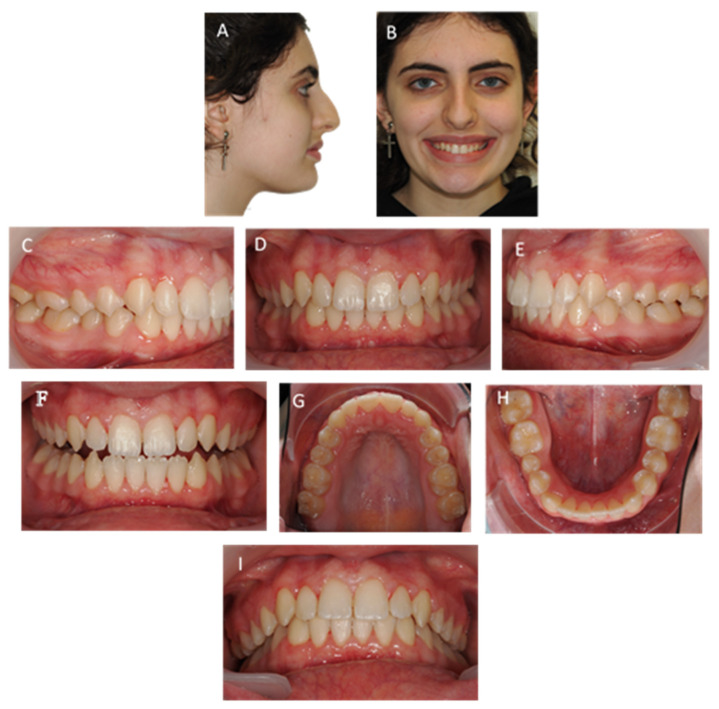
Final photographs, 3 years after the treatment began, including the use of aligners at night; (**A**) profile photo; (**B**) smile; (**C**–**E**) intra-oral photos in maximum intercuspation/centric relation; (**F**) protrusive guide; (**G**,**H**) occlusal photos; (**I**) overjet view.

**Figure 14 children-10-00624-f014:**
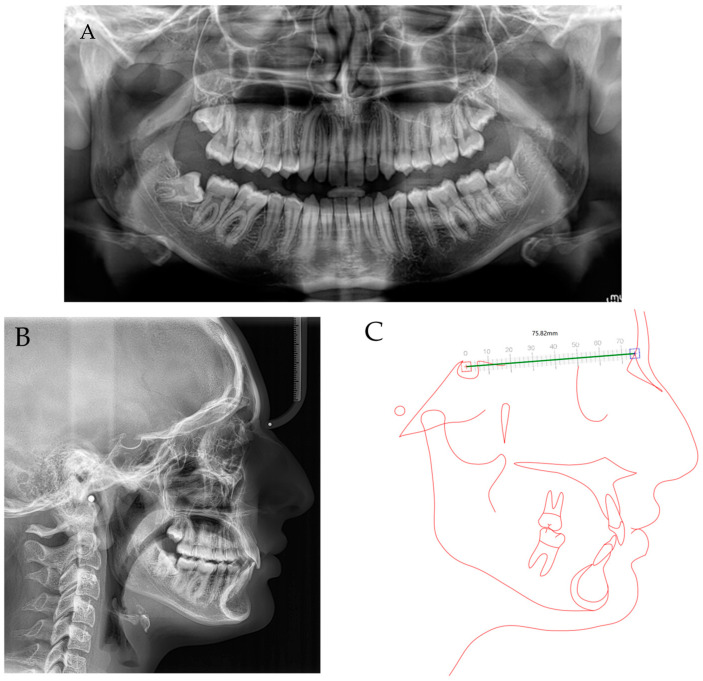
Final panoramic X-ray (**A**), teleradiograph (**B**) and cephalometry (**C**).

**Figure 15 children-10-00624-f015:**
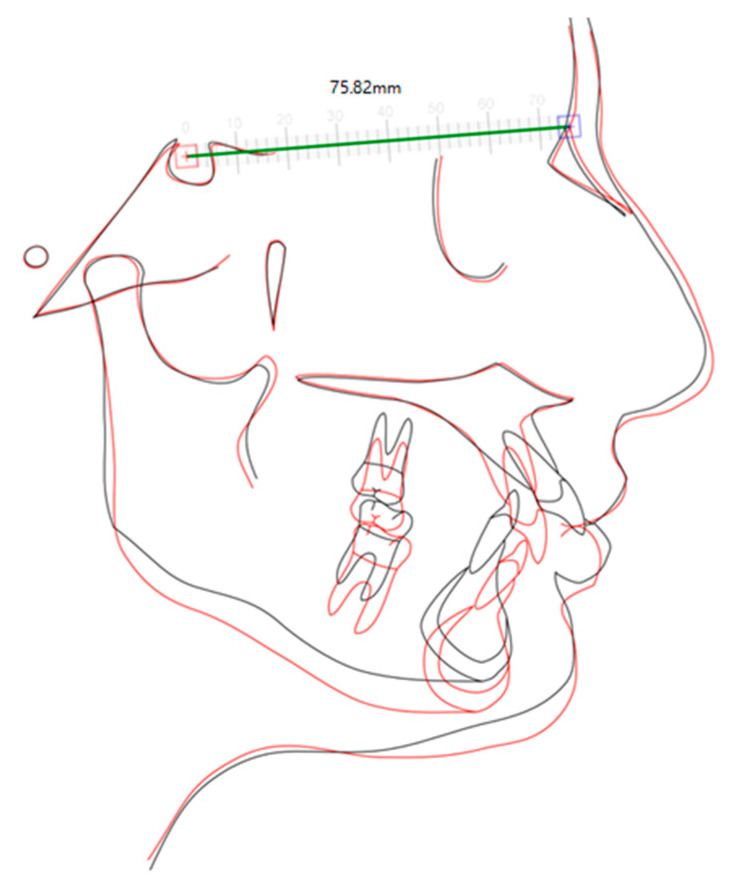
Superimposition of the initial tracing (black) and final tracing (red).

**Figure 16 children-10-00624-f016:**
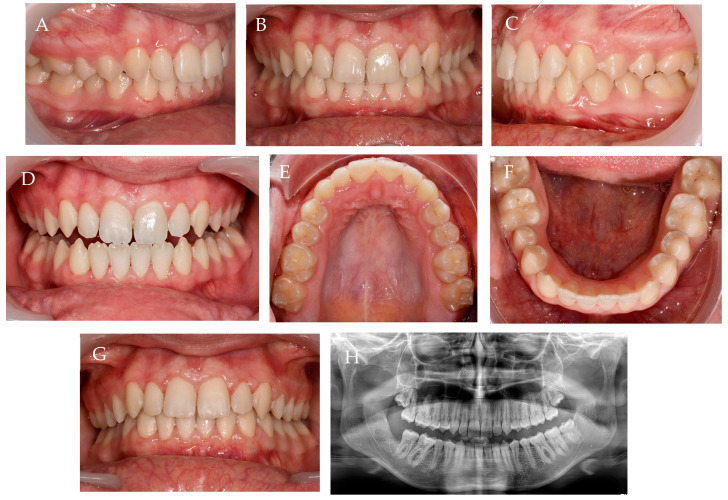
Follow-up, 6 months after treatment end. (**A**–**C**) intra-oral photos in maximum intercuspation/centric relation; (**D**) protrusive guide; (**E**,**F**) occlusal photos; (**G**) overjet view; (**H**) panoramic X-ray.

## Data Availability

Not applicable.

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
