# Peer review of "Scissor Bite in Growing Patients: Case Report Treated with Clear Aligners"

_children, 2023, doi:10.3390/children10040624_

Round 1

Reviewer 1 Report (Previous Reviewer 3)

Dear Editor

the case have been treated to an almost excellent standard.

Mixing a case report with a literature review carried out in a sort of hybrid fashion does not seem to me a good idea. The findings of the so called review can be used however in the discussion section.

The article is case report and mixing it with a literature review is of limited significance.

I would avoid to report any sharp conclusion from a single case report.

Author Response

First of all, we would like to thank the commentaries and suggestions. We have proceeded to the modification of the literature review component, as we moved it to the discussion section, with table 1 (old table 2) attached as supplementary materials.

Also, we have changed our conclusion, focusing on our case report, as requested.

Reviewer 2 Report (Previous Reviewer 2)

Dear authors,

The revised version of the manuscript is improved, however nothing changed regarding methodology and quality of presentation. There are multiple issues that need to be addressed prior to publication. Please see the attachment.

Author Response

We have changed now the literature review section, to discussion.

Also, we have made the suggested modifications on the manuscript file. About the mentioned topics:

  • We have updated our search period until January 2023. Although, the articles found didn’t meet the inclusion criteria of our literature review.
  • We have previously sent the patient informed consent via e-mail as requested and we will also sent it again in this resubmission.
  • We avoided using personal terms.
  • We have removed the old table 1, discussing the literature review only on discussion to supplement our case report.
  • We have made all the other modifications requested in the pdf that was sent.
  • We have adapted the previous “Analysis of the Results obtained in literature review” section, to the discussion.
  • We have decided to put the following sentence in Discussion “For the literature review, the following keywords and MeSH terms were employed in the search strategy, in the PubMed and VHS database, from 2002 to January 2023: "scissor bite OR brodie bite" AND "malocclusion" AND "treatment OR correction OR therapeutics" nnly to explain what we have done with the literature review. We didn’t add a Material and Methods section only to not collide with our type of article – case report. Then, we focus on removing the literature review tables to supplementary materials, discussing it in the Discussion section.

Thank you very much for your support, and hope our modifications are enough for the acceptance of our article.

If you need something more, please let us know.

Round 2

Reviewer 1 Report (Previous Reviewer 3)

Dear authors

focusing only on the case report improved the article grately

best regards

Author Response

We would like to thank the reviewer for this commentary.

Reviewer 2 Report (Previous Reviewer 2)

Dear authors,

There are still multiple issues that need to be addressed prior to publication:

First and foremost, there are multiple grammar and spelling errors, English needs to be revised by a professional.

Second, although several changes were performed in the methodology there are multiple issues that require correction. Please see the attachment.

Author Response

In the last resubmission, we have already used EditMyEnglish platform to correct the grammar and spelling errors. We have sent the certificate via e-mail to the editorial office. Although, we send it again.

We have made all improvements suggested in this last attachment, and we send it now with all the modifications in yellow (the new ones).

Thank you very much for your support, and hope our modifications are enough for the acceptance of our article.

If you need something more, please let us know.

This manuscript is a resubmission of an earlier submission. The following is a list of the peer review reports and author responses from that submission.

Round 1

Reviewer 1 Report

The format of this manuscript does not follow general format of scientific articles. The systematic review was not performed properly nor the case series report. I would like to separate the case report and the systematic review. 

Reviewer 2 Report

Dear Authors,

Congratulations on the work you have done and presented in this paper. You will see that your work can be published only after some major modification:

1. First and foremost there are multiple issues regarding spelling, text style is inconsistent, there are multiple grammar errors, reference style is not consistent with the journal requirements. References section needs to be improved, 21 references are not enough for a systematic review.

2. The methodology of your manuscript is by far the weakest part of the article, please see the attachment.

Reviewer 3 Report

Dear Authors

The PICO question is not applicable in your case since you are not comparing interventions please reconsider using a PIO or PEO approach

A specific question should be formulated and this is not your case 

The given question was not used in the search, etiology is in the question but not in the search string

The search performed in Pubmed with the enclosed string delivered 77 items and not the 55 in the flow chart, only three articles were published after march 2022

In the flow chart the reasons for exclusion of the article whose full text was assessed should be given

The review was not registered in any database (Prospero or OSF)

The case reported was very difficult and the result is very good but the level of finishing is not adequate for its publication in a high impact journal

Reviewer 4 Report

It was my pleasure to review the manuscript entitled (Scissor bite in growing patients: systematic review and a case 2 treated with Clear Aligners with Mandibular Advancement; ID: children-2120099) submitted to the Children journal. The study aim was to assess the orthodontic treatment approaches of children with scissors bite. The study has critical and severe flaws in research methodology. The study did not adhere to the PRISMA guidelines for conducting systematic review. Moreover, it lacks proper scientific presentation and data interpretation.